# Fake news stance detection using selective features and FakeNET

**Turki Aljrees**[1], **Xiaochun Cheng**[2], **Mian Muhammad Ahmed**[3], **Muhammad Umer**[3]*,
**Rizwan Majeed**[4], **Khaled Alnowaiser**[5], **Nihal Abuzinadah**[6], **Imran Ashraf**[7]*

**1** College of Computer Science and Engineering, University of Hafr Al-Batin, Hafar Al-Batin, Saudi Arabia,
**2** Department of Computer Science, Swansea University, Bay Campus, Swansea, United Kingdom,
**3** Department of Computer Science & Information Technology, The Islamia University of Bahawalpur,
Bahawalpur, Pakistan, **4** Faculty of Computer Science and Information Technology, Universiti Tun Husein
Onn Malaysia (UTHM), Bahru, Malaysia, **5** Department of Computer Engineering, College of Computer
Engineering and Sciences, Prince Sattam Bin Abdulaziz University, Al-Kharj, Saudi Arabia, **6** Faculty of
Computer Science and Information Technology King Abdulaziz University, Jeddah, KSA, **7** Department of
Information and Communication Engineering, Yeungnam University, Gyeongsan, Republic of Korea

* umersabir1996@gmail.com (MU); imranashraf@ynu.ac.kr (IA)

## Abstract

The proliferation of fake news has severe effects on society and individuals on multiple
fronts. With fast-paced online content generation, has come the challenging problem of fake
news content. Consequently, automated systems to make a timely judgment of fake news
have become the need of the hour. The performance of such systems heavily relies on fea-
ture engineering and requires an appropriate feature set to increase performance and
robustness. In this context, this study employs two methods for reducing the number of fea-
ture dimensions including Chi-square and principal component analysis (PCA). These meth-
ods are employed with a hybrid neural network architecture of convolutional neural network
(CNN) and long short-term memory (LSTM) model called FakeNET. The use of PCA and
Chi-square aims at utilizing appropriate feature vectors for better performance and lower
computational complexity. A multi-class dataset is used comprising 'agree', 'disagree', 'dis-
cuss', and 'unrelated' classes obtained from the Fake News Challenges (FNC) website. Fur-
ther contextual features for identifying bogus news are obtained through PCA and Chi-
Square, which are given nonlinear characteristics. The purpose of this study is to locate the
article's perspective concerning the headline. The proposed approach yields gains of 0.04
in accuracy and 0.20 in the F1 score, respectively. As per the experimental results, PCA
achieves a higher accuracy of 0.978 than both Chi-square and state-of-the-art approaches.

## Introduction

Data is being produced online at an unprecedented rate in this technological era. Unfortu-
nately, an enormous percentage of fake news is flooded over the internet. Fake news is created
to attract the audience, influence individual decisions, and play with people's beliefs [1–3].
Such content is generated and uploaded to raise the cash which is produced by clicking on
such content [4]. This content is also used to affect key events such as political elections, social

org/10.1371/journal.pone.0287298

EGYPT

**Data Availability Statement:** Data used in this
study is publicly available on the following links. 1.
http://www.fakenewschallenge.org/ 2. https://www.
kaggle.com/datasets/abhinavkrjha/fake-news-
challenge.

**Funding:** No: The author(s) received no specific funding for this work.

**Competing interests:** The authors declare that there is no conflict of interests.

campaigns, etc. [5]. This is achieved by deliberately misleading the readership through fake or altered content. Due to the ease with which information can be obtained and disseminated via social media, detecting fake news based only on its substance is a challenging and nontrivial task. Some studies show, for instance, that Russia has used social bots and phony accounts to disseminate false information. A recent survey found that 66% of American adults believe that false news creates "a significant deal of confusion" regarding the truthfulness of the news [6]. In addition, the business, marketing, and stock-share sectors are witnessing the escalating negative effects of a widespread misleading information cascade. In 2013, for instance, fake news propagated on Twitter that Barack Obama was injured in an explosion, wiping away 130 billion dollars in stock value [5]. There have been claims that fake news had a significant role in the rise of political polarization and party conflict throughout the 2016 US presidential campaign, as well as its outcome [7–9]. Therefore, it is obvious that identifying false news is a major problem for the media, and that methods to identify such stories are essential.

Because manual checking requires longer time and laborious effort, the natural language processing (NLP) community has shown increasing interest in the automatic identification of fake news [10, 11]. However, even automatic systems find it difficult to determine whether or not a news article is credible [12]. One possible initial step in detecting fake news is to compare the article in question to coverage of the same issue in other media outlets which is called stance detection. Several jobs rely on stance detection as a basis including the analysis of online discussions [13–15], verification of Twitter rumors [16, 17], and the comprehension of the logical progression of an argumentative article [18].

Pomerleau and Rao (2017) hosted the first False News Challenge (FNC-1) [19] to stimulate the development of automated fake news detection techniques. Such techniques involve employing artificial intelligence (AI) technology and machine learning. It was estimated that 50 teams from both the business world and universities took part in this competition. The FNC-1 challenge asked participants to identify the article's perspective in light of a predetermined title. An article's stance can fall into one of four categories. It may support, refute, or otherwise address the claims made in the headline. The FNC-1 task's guidelines, dataset, and evaluation measures may all be found on the official website [19]. In Table 1 we can see four documents that illustrate various points of view.

Different types of deep learning models [20–23] have gained a lot of popularity in NLP tasks such as question answering [24, 25], finding the semantic similarity in texts [26, 27], text analysis [28, 29], etc. The semantic similarity of the two questions is calculated using Siamese MaLSTM in [30]. In this case, the relevance of each headline-body pair is being evaluated [9, 31–33] by using the word embedding features with supervised learning models.

In this research, we propose a technique that can automatically assign labels such as 'agree', 'disagree', 'not relevant', and 'discuss' to news stories. The degree of concordance between the headline and the allocated body is used to determine the classification. Articles of interest can

**Table 1. Sample from FNC dataset and respective stances.**

| Headline | Hundreds of Palestinians flee floods in Gaza as Israel opens dams. |
|---|---|
| Agree | Hundreds of Palestinians were evacuated from their homes Sunday morning after Israeli authorities opened a number of dams near the border, flooding the Gaza Valley in the wake of a recent severe winter storm [..]. |
| Disagree | Hundreds of Palestinians were evacuated from their homes Sunday morning due to severe winter storm[..]. |
| Discuss | Palestine accuses Israel of opening dams, flooding Gaza, forcing evacuations[..]. |
| Unrelated | It's 'rubbish' that Robert Plant turned down £500m Led Zeppelin reformation offer, says publicist[..]. |

be located by looking for relevant keywords in article titles, which is the basis of the proposed methodology. Some of the phrases used in the headlines might be used to locate pivotal paragraphs in the main text. The data in Table 1 reveal that only the first paragraph of the main body is directly related to the headline. In the FakeNET model, the feature set is sent to the embedding layer, both with and without preprocessing, to be transformed into word vectors. To undertake component-level analysis and acquire the reduced feature set, the second round of experiments is conducted using Chi-square and principal component analysis (PCA) features.

PCA is a widely used statistical method for selecting features [34]. Using PCA, the classifiers' ability to distinguish between similar examples is improved. Among its many uses are face recognition, text classification, and image compression [34]. The essence of PCA is to reduce variables to a smaller number with the highest correlation [35]. PCA is a useful tool in many fields due to its ability to reduce the dimensionality of a feature set by a linear transformation. The simplified dataset is nevertheless consistent with the original in many ways [36–38]. The number of characteristics in the new dataset may be higher or lower than in the old one. Finding the principle components requires using the covariance matrix. Once the features have been collected via one of the aforementioned techniques, they are sent to the embedding layer of a deep learning model.

This research work analyzes the impact of feature reduction techniques with the ensemble of two popular learning models CNN and LSTM called FakeNET. The effectiveness of Fake-NET is the improvement of 0.20 in F-score and 0.04 accuracy than the previous state-of-the-art research.

The remaining paper is organized as follows. State-of-the-art works closely related to this work are discussed in Section 2. The details of the dataset, preprocessing steps, model explanations, and model parameters are described in Section 3. The results of extensive experiments with discussion are presented in Section 4. Section 5 describes the conclusion of the paper and future work.

## Related work

In NLP, stance detection is a common and extensively studied task. The phrase refers to the process of reading between the lines of a text to ascertain the reader's attitude toward the subject at hand [39]. Many other tasks, such as detecting bogus news [19], validating claims [40], and searching for arguments [41] rely on stance detection. Earlier research on detecting false news focused on target-specific attitude prediction where the opinion of a text item was predicted about a topic or a named entity. Many studies [39, 42, 43] have used target-specific stance forecasting for tweets and online arguments. Structure [13], language, and lexicon [15] are the foundations of such context- and audience-aware methods.

Tweet and online discussion stance prediction differ from news article stance identification, where the latter is NLP-based and contextualized by the article's headline. Claims' veracity is predicted by factoring in the articles' bias and the trustworthiness of their sources [40]. Stance traits are used to determine the veracity of false news, which are themselves defined as 'unsupported statements' [44]. To determine the veracity of tweets, one researcher employed the hidden Markov model with simply the tweet's publication time and positions as features [45]. A solution to the claim relevancy finding the problem is provided in another study [46] which uses a variety of machine learning or information retrieval to achieve a 91.6% success rate.

In 2017, the first competition to determine false news's bias position was launched. The work proposed in [47] to categorize the perspective of one sentence of a headline story against a particular claim served as inspiration for the FNC-1 perspective-detecting task. FNC-1

challenge employed a dataset based on the developing dataset [47] which was partially labeled. At the document level in FNC-1, the position is identified by categorizing the entire news item about the headline. The SWEN [48] system built by the Talos Research Intelligence team's SOLAT ranks first among all FNC-1 systems. It utilizes deep CNNs, pre-trained vectors from Google News, and gradient-boosted decision trees with a weighted ensemble average of 50/50 approach. There was an overall accuracy of 82.02% for the model.

The 1$^{st}$ runner-up 'Athene team from 'Technische universität Darmstadt (TU Darmstadt)' employed a multi-layer perceptron (MLP) as just a combination of 6 layers with artisanal details [49] to achieve an accuracy of 81.97%. The team comprised researchers from the Commonplace The Heterogeneous Publications Research Training organization's information lab processes and adapts information. The second place UCL machine reading (UCLMR) presented an MLP model using term frequency (TF), bags of words (BoW), and TF-inverse document frequency (TF-IDF) as features and achieved an accuracy of 81.72% [50]. The runner-up team uses both semantic embeddings and lexical matching to extract features which are then fed into a different set of gradient-boosting trees. In addition, a classifier based on two rounds of logistic regression [4] and an ensemble of five classifiers was [51] placed ninth and eleventh, respectively. Classification-based algorithms [52] use both manually produced features and features generated by neural networks, as used by the three challenge winners [48–50] in FNC or semEval.

Another study looked into how an agreement-aware article search could be used to predict the spread of rumors. To give consumers a more complete picture of a topic for which the underlying truth is uncertain, they created an agreement-aware search framework. The authors created a two-stage model that consists of a tree-based model with manually-crafted attributes or an RNN plus attention model that zeroes in on a select few lines [53]. In [52], a model is proposed that uses TF-IDF to extract features that may be utilized to display both headlines as well as bodies of news items using a single, MLP-based, end-to-end ranking algorithm. After training on FNC-1, the model achieves an accuracy of 86.66% percent.

In [12], the stance detection issue from the FNC-1 task is tackled using a deep learning approach. It uses bidirectional RNNs, max-pooling, and neural attention mechanisms to construct representations from news item titles and bodies, then combines those representations that resemble anything outside of themselves. Pre-training is used to combine brain representations with an exterior resemblance characteristic, and the resulting accuracy of 83.8% is impressive. In contrast, [9] employs a deep recurrent model to calculate the statistical features which are calculated using a weighted n-gram BoW model, and the manually created external features are extracted using feature engineering techniques. Finally, a deep neural layer is used to aggregate all the information and classify pairs of news headlines and bodies as agree, disagree, discuss, or unrelated. The results show that an accuracy of 89.29% is obtained.

The superior performance of a neural network over manually constructed features is demonstrated in [32]. The model's 86.5% accuracy was accomplished through the use of the addition of a technique for dividing the current attentive reader's full attention between content in the body and headlines, and the bilateral multi-perspective matching model (BiMPM). An LSTM model with attention received an 80.8% F score in [33]. Similarly, [31] presents the use of a conditioned bidirectional LSTM with global characteristics. The results show that 87.4% accuracy is achieved when combining global features with local word embedding features to predict the stance of headline-article combinations. In [52], the authors employ a ranking-based approach to the stance identification problem rather than a classification-based one. The ranking-based approach evaluates two sets of headlines and bodies and seeks to find the largest possible gap between their respective true and false positions. With this method, accuracy is increased to 86.66%.

The Normalised Difference Vegetation Index was used to assess the computational expense using the Random Forest classification algorithm [54]. It has been shown in the past that a number of variables limit the accuracy of ML models, the author employed a CNN-based model to develop an automated weed detection system [55]. Deep learning models have been used in many types of research like air pollution forecasting [56], image classification [57], botnet detection [58], Botnet attack detection [59], and intrusion detection [59]. Authors employed LSTM to observe the change in groundwater storage [60], and climate change forecasting [61].

Recently, novel stacked CNNs were introduced in [62] and an innovative method based on the introduction of stacked Bi-LSTM layers in [63]. For modeling sequences, the LSTM layer is employed. Bi-LSTM is superior because it takes into account context from both ends of the sentence. Several models are used to analyze FNC-1 data in [64] end-to-end memory networks, CNN, LSTM, and a mixture of the two. The authors also suggest a new addition to the overall architecture which relies on a matrix of similarities. They conclude that sMemNN with TF has a maximum accuracy of 0.885. CNN + LSTM and LSTM + CNN, on the other hand, only manage 0.485 and 0.653 accuracy, respectively. The reason is the distribution of data regarding training and testing. When training, it is important to have a fair distribution of data across classes, so that each epoch contains around the same number of examples. In addition, the CNN architecture does not have a pooling layer, which could account for the subpar results. The study [65] used RoBERTa transformer language model for a comprehensive attitude language model identification. On the FNC-I benchmark dataset, the model scored 93.71 percent accuracy. The aforementioned studies that use machine learning models rely on manually constructed characteristics. Because of their inability to consider the overall context of the text, the results from using these features are often subpar. Most of the models also fail to provide satisfactory agreement and disagreement class detection performance. FakeNET uses PCA and Chi-square testing in conjunction with CNN and LSTM layers to circumvent these restrictions. In comparison to previously described deep learning strategies, the proposed approach achieves a higher rate of success.

## Materials and methods

This study proposes an approach for fake news stance detection. The flow of the adopted methodology is shown in Fig 1. Starting with the data acquisition, the methodology follows feature reduction using PCA and Chi-square. The extracted features are fed into the proposed hybrid model which comprises CNN and LSTM. The data split ratio for training and testing is 0.7 to 0.3, respectively. The model is evaluated using accuracy, precision, recall, and F-score.

### Dataset

The dataset for experiments is obtained from the official website [66]. The FNC dataset consists of 75385 tagged instances and 2587 article bodies, which roughly correspond to 300 headlines. There are 5 to 20 news articles for each allegation. Table 2 shows that of these headlines, 0.074 are agreed upon, 0.020 are disagreed upon, 0.177 are discussed, and 0.728 are unconnected. Manual labeling is used for the assertions about the body of the articles. The labels' specifics are as follows:

- The label 'agree' depicts that the article body and headlines are related to each other.

- The label 'disagree' depicts that the article body and headlines are having no relation.

- The label 'discuss' depicts that there is some type of similarities between the article body and headlines.

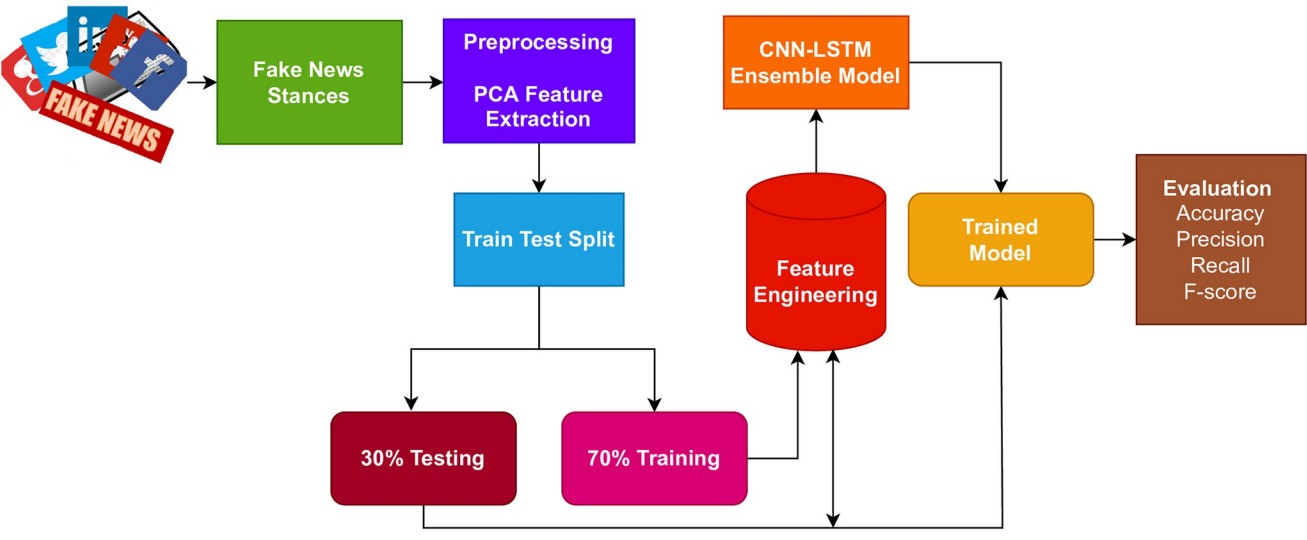

**Fig 1. Flow diagram of the adopted methodology.**

- The label 'unrelated' depicts that the article body and headlines are totally different concerning context.

As the FNC dataset is a benchmark dataset and there are some rules for using this dataset. The rules included that the best-performing model is considered only if it has a 49972 number of samples as a training set and 25413 as a testing set. In this division of the training-testing set, the number of headlines and articles bodies in the training set is 1648 and 1683 and in the testing set, they are 880 and 904 respectively.

## Data preprocessing

In data mining, preprocessing is the process of cleaning and standardizing raw data before it is processed further. The FNC-1 dataset has gone through several text preprocessing procedures. Applying algorithms from the Keras toolkit, we were able to complete these tasks using NLP methods like casing-shifting, stopword elimination, stemming, and tokenization.

Stopwords, such as 'of', 'the', 'and', 'an', and so on are examples of common words found in the text that contribute little to the text's qualities and are therefore unnecessary here. By excluding the aforementioned pointless words, we save time processing and storage space. Words with comparable meanings (e.g., game and games) might appear more than once in the text. If that is the case, simplifying the language to a single universal language is powerful. This is done using the Porter stemmer algorithm from the NLTK.

Following the above-described pre-processing methods, there were 372 fewer terms in the headlines. Each headline was parsed into a vector using the tokenizer function in the Keras

**Table 2. Fake news challenge dataset details.**

| headlines | tokens | instances | agree | disagree | discuss | unrelated |
|---|---|---|---|---|---|---|
| 2587 | 372 | 75385 | 0.074 | 0.020 | 0.177 | 0.728 |

toolkit. After the completion of these steps, we utilized word embedding (word2vec) to convert the text into a vector list. In the end, 5000 unigrams taken from article titles and bodies are compiled into a dictionary. All of the headlines will be the maximum length allowed. Headlines that are too short to meet the maximum duration requirement are padded with zero. The features are then given to a CNN [67] and LSTM layer hybrid neural network design.

## Approaches to reduce dimensionality

When it comes to text classification, feature extraction, and feature selection are the two methods of dimensionality reduction. Feature selection techniques involve keeping only the most important and relevant features and discarding the rest [68]. Via contrast, the vector space is transformed in feature extraction methods to produce a new vector space with unique properties [34]. A new vector space is created in which the characteristics are decreased.

Performance increases as a result of reduced processing time due to feature reduction [69]. Improving text classification accuracy is very sensitive to feature reduction [69]. Because of this, picking the proper selection algorithm to cut down on dimensions is of the utmost importance. Several popular feature reduction approaches exist including PCA, mutual information [70], Gini coefficient (GI), and Chi-Square statistics, etc. The text classifier's scalability can be enhanced by combining deep learning models with techniques for decreasing two-dimensionality including PCA and Chi-square.

**Principal component analysis.**   Popular for its ability to decrease the dimensionality of attributes set by a linear transformation, PCA is a useful tool in many fields. The resulting data set has been streamlined for readability while preserving the essential features of the original [37]. The number of characteristics in the new dataset may be higher or lower than in the old one. When calculating PCA, the covariance matrix is referred to. These parts are listed from most important to least [71]. The following equation describes the transformation of the original matrix, which we suppose has $a$ dimensions and $b$ observations, into a $t$-dimensional subspace.

$$Y = (E^Z X) \qquad (1)$$

In the equation, $E_{a \times t}$ is the projection matrix. It includes $t$ eigenvectors that correspond to $t$ of the greatest possible eigenvalues. $X_{a \times b}$ seems to be the mean-centered data matrix. In this research work, we have used an optimized version of PCA known as robust PCA for the extraction of significant features using the feature reduction technique. While selecting the optimized number of PCA components is done using the 'Scree' plot.

**Chi-square.**   Chi-square is among the best algorithms for feature selection [72]. It is built for examining hypotheses involving pairs of discrete categories. Both the degree of extremeness and the degree of dependence between $a$ and $b$ can be estimated using this method and compared to the chi-square distribution with one degree of freedom [69, 73]. Chi-square is used for the independence test and goodness-of-fit test. It uses a test of independence to determine whether or not a feature is dependent on the target label to make a selection (s). Chi-square tests the consistency between variables. The features with a positive correlation are retained, while those without are discarded. Chi-square tests are performed on each feature individually concerning the target class, and the significance of each feature is determined by comparison to a fixed threshold (which is 0.05 commonly). For a given feature, the larger the chi-square value, the less significant it is. A similar trend holds for chi-square, where a smaller number indicates a higher level of importance. Chi-square has been used by many studies to be an effective tool for text classification with fewer features [72, 74]. Eq 2 shows the chi-square

feature selection formula

$$X_c^2 = \Sigma \frac{(O_i - E_i)^2}{E_i} \qquad (2)$$

where *c* represents the degree of freedom, *O* represents the observed value, *E* represents the predicted value, and $X^2$ represents the chi-square computed value for the feature.

## Proposed model

This study contributes a hybrid deep learning model that combines two types of neural network layers CNN and LSTM with a feature reduction technique. The suggested method outperforms conventional deep learning models in terms of prediction. Four data models are created to examine the connection. The first model employs categorization directly from the raw data without any preparation of the characteristics. After preprocessing, the second model makes use of the whole set of features without any reductions. Dimensionality reduction [75, 76] methods like PCA and Chi-Square testing are used to create the third and fourth models. In this study, we dig deeper into the question of which models function best with the hybrid CNN/LSTM model while processing textual input.

The FakeNET architecture receives the features that have been picked by any of the aforementioned four models. The input headlines and article bodies are passed to the model's embedding layer, where each word is assigned a 100-dimensional vector representation. Given that there are 5000 features, this layer produces a matrix with those dimensions. With the weights obtained via matrix multiplication, we can generate a vector for each word, which is stored in the output matrix. To extract contextual characteristics, these vectors are sent to a CNN layer. To generate a single stance as the final output, the CNN layer's output is transferred to an LSTM layer, which in turn is passed to a fully linked dense layer. As can be seen in Fig 2, the proposed model is trained and tested using 32-sample subsets of the entire dataset.

There are *w* entries in the dataset's text sequence *a*. A dense vector with dimension *d* is used to represent each *w*. Input *a* feature map has $d \times w$ dimensions. The first step is to use the Keras tokenizer to turn the headlines and main text into tokens. Then, the tokens are turned

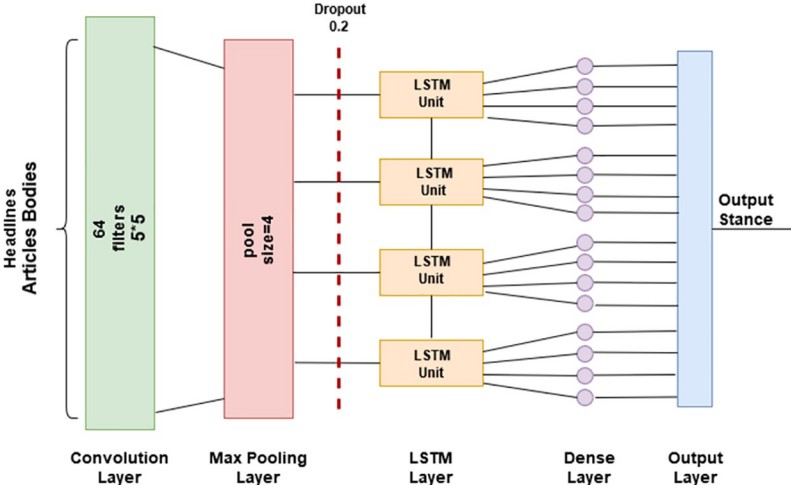

**Fig 2. The architecture of the proposed model.**

into word vectors using word2vec word embedding in the Keras embedding layer. The first and second models both use the word vectors output the convolution layer receives its input from the word embedding layer. On the other hand, PCA and Chi-squared are used to obtain relevant attributes before applying them to models three and four. An embedding layer then takes these features and turns them into word vectors. The word vectors are then sent to the convolution layer. Convolution layers are used to learn semantic or structural features. $n$ CNN neurons receive a word vector as input. Filters of varying diameters allow us to obtain a wide range of characteristics. Multiple filters $f$ of different kernel sizes $c$ are applied on each word embedding $e$ and the output is generated as $(c \times e)$. The current study uses a 5-word kernel, therefore a 64-word filter generates 5-word combinations. The FakeNET model comprises a convolutional layer with 64 filters of 5 size. It is followed by the max pooling layer with a 4-pool size. Each LSTM layer has 100 neurons while the dense layer has 50 neurons. The dropout layer is used with a 0.2 dropout rate. This study used rectified linear unit (ReLU) as the activation.

ReLu's activation layer is used to demonstrate the network's non-linearity and to normalize all negative values to zero. Since the function has no bearing on the CNN layer's outcome shape, the latter is identical to the former. After being processed by the ReLu activation function, the value of each neuron is passed on a 1-dimensional maximum pooling layer. With this layer, the input from all kernel sizes is combined into a single output by taking the maximum from each kernel. Overfitting can be avoided and the size of input characteristics for subsequent layers can be drastically reduced. The pool size $p$ is 4, therefore this layer's output further reduces the features. Overfitting can also be mitigated using the dropout layer, which discards input values below a predetermined threshold called the dropout rate. Since no value in the FNC-1 dataset is less than 0.2, the dropout layer's output is identical to the input.

The LSTM layer is used with 100 units. For the data to be useful, it must be generated in a chain-like sequence with the history of inputs preserved. Due to its three-part structure, input gate $i_k$ forgot gate $f_k$, and output gate LSTM is ideal for this task. With the dropout value in mind, these gates determine which pieces of data are crucial for classification and which may be safely ignored.

As the last component of the proposed architecture, a dense layer with all connections generates a single output. The softmax activation function comes after this layer. The softmax function is used to classify multiple categories. The dataset used in this study has four classes, thus we utilized softmax activation and employed the 'Adam' optimizer. In our tests, we use 32 batch size and 50 epoch iterations. The complete layer-wise details and each layer hyperparameters values of the proposed model are shown in Table 3.

**Table 3. Layer-wise hyperparameters details of the proposed FakeNET model.**

| CNN-LSTM |
| --- |
| Conv (5, @64), activation='relu' |
| Max Pooling (4 × 4) |
| LSTM (100 neurons) |
| Dropout(0.2)Dense (64 neurons) |
| Dropout(0.2) |
| Dense (32 neurons) |
| Dense (4 neurons) |
| Softmax (4-class) |

## Performance evaluation metrics

This study utilizes measures of accuracy (A), precision (P), recall (R), and F-score (F) to assess the model's performance and compare its efficacy. Accuracy is the rate of correctly classified predictions. Precision tells us about the factualness of the model. Recall provides information about the model's comprehensiveness. The F-score metric demonstrates the fullness of the recommended model in terms of class-wise accuracy.

$$A = \frac{True\ Positive + True\ Negative}{True\ Positive + True\ Negative + False\ Positive + False\ Negative} \tag{3}$$

$$P = \frac{True\ Positive}{True\ Positive + False\ Positive} \tag{4}$$

Precision is calculated as the ratio of correctly classified positive class and the sum of correctly and falsely classified values of the positive class. It tells us about the factualness of the model.

$$R = \frac{TruePositive}{TruePositive + FalseNegative} \tag{5}$$

The recall rate is determined by dividing the number of instances in which a positive class was assigned a value by the total number of instances in which a positive class was assigned a value and a negative class was assigned a value. It provides information about the model's comprehensiveness.

$$F1 = 2* \frac{\cdot precision \cdot recall}{precision + recall} \tag{6}$$

The F1 score evaluates the performance of the model for each category. When the data is unbalanced, the F-score measure is commonly utilized. FNC-1's dataset is similarly very imbalanced, hence we utilize As a measurement, the F1 score metric to demonstrate the fullness of the recommended model in terms of class-wise accuracy.

## Results and discussion

Experiments involve the use of 25413 headlines and articles to test the proposed FakeNET ensemble model which is trained on 49972 samples. A 2 *GB* Dell PowerEdge *T*430 GPU running on a 2*x* Intel Xeon 8 Core, 2.4*Ghz* system with 32 GB of DDR4 RAM is used for the training. Using learned embeddings, the training on the FNS-1 dataset takes 15 minutes and the classification results are displayed at the end of the process. Conversely, the computation time for feature reduction methods is 2 minutes.

### Results of different models

While the LSTM's ability to process sequential information is well-known, generating sequences from a big dataset can be a time-consuming process that can lead to an overfit. However, CNN lacks a memory unit, hence it is unable to process data in sequences. So making an ensemble of CNN and LSTM is a good option to obtain better results. We examine the results obtained by a FakeNET architecture using a reduced feature set, PCA, and chi-square. Fig 3a and 3b depict train and test accuracy and loss,

Table 4 shows experimental results regarding four variations used in this study. It is clear from the results that PCA is the way to go when dimension reduction is required for a large

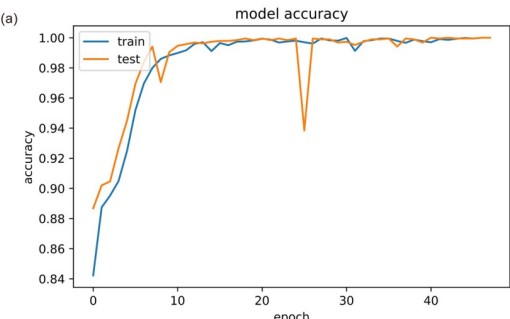
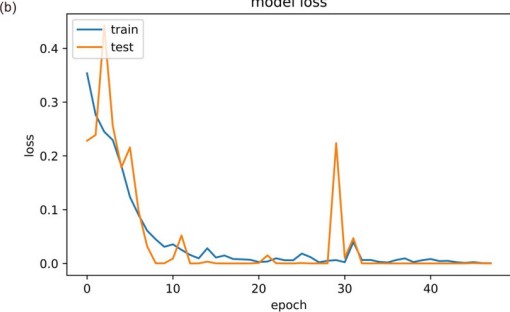

**Fig 3. Training and testing curves of proposed model for, (a) Accuracy, and (b) Loss.**

feature set. Results show that the presented model has the highest accuracy than other variants. The FakeNET achieves the best accuracy score of 0.978 when trained using PCA-based features. Similarly, it gains the best scores of 0.974, 0.982, and 0.978 for precision, recall, and F-score, respectively.

From the outcomes, it is clear that when the features are employed accuracy is low without any preparation or data cleaning. It suggests that the original dataset has inconsistencies, duplications, and noise. The accuracy increases considerably when the preprocessing steps are carried out and unnecessary data is removed, reaching an accuracy score of 0.930. Additionally, Chi-Square improves accuracy to 0.950 by selecting important features from the data.

## Comparison with deep learning models

The results of the FakeNET are compared with other deep learning models as well. For this purpose, bidirectional encoder representations from Transformers (BERT), XLNet, and RoBERTa are used.

All parameters are fine-tuned in the pre-trained model and a basic classification layer is added for the results reported on the FNC-1 task using the BERT [77]. BERT makes predictions for every masked location on its own. It does not consider inter-dependencies between anticipated masked positions while learning.

When used together, the bidirectional context and the elimination of independent predictions [78] make XLNet a powerful tool. Instead of predicting tokens in sequential order, it uses the 'permutation language modeling' technique to predict characters randomly. Using transformer XL as its base architecture, XLNet outperforms BERT on 20 different tasks.

Robustly optimized BERT approach (RoBERTa) is a free and open-source language model released in July 2019 [79]. Transfer learning on a 12-layer, 768 hidden unit, 12 attention heads, deep transformer model based on RoBERTa, and a total of 125M parameters, the author of [77] builds a large-scale language model. Training for a total of 50 epochs and using the

**Table 4. Classification result of FakeNET model.**

| Model Name | A | P | R | F |
|---|---|---|---|---|
| FakeNET + without preprocessing | 0.784 | 0.814 | 0.824 | 0.819 |
| FakeNET + preprocessing | 0.930 | 0.960 | 0.970 | 0.960 |
| FakeNET + Chi-square | 0.952 | 0.923 | 0.911 | 0.914 |
| **FakeNET + PCA** | 0.978 | 0.974 | 0.982 | 0.978 |

**Table 5. Performance comparison of deep learning approaches with FakeNET.**

| Model Name | A | F | F-agree | F-disagree | F-discuss | F-unrelated |
|---|---|---|---|---|---|---|
| BERT | 0.913 | 0.728 | 0.647 | 0.478 | 0.800 | 0.986 |
| XLNet | 0.921 | 0.760 | 0.686 | 0.548 | 0.821 | 0.845 |
| RoBERTa | 0.937 | 0.781 | 0.707 | 0.580 | 0.845 | 0.991 |
| FakeNET | 0.978 | 0.978 | 0.860 | 0.760 | 0.880 | 0.990 |

hyperparameter recommendations provided by [79] this study obtains the results from RoBERTa and compares it with both BERT and XLnet models.

Table 5 shows the comparison of the results for BERT, XLNet, RoBERTa, and the proposed model. On the FNC-1 challenge, BERT achieves 0.913% accuracy while the F-score of BERT is significantly low. Class-wise results show that it obtains the highest F-score for the 'unrelated' class which is 0.986. XLNet shows better results than BERT and attaining 0.760 F-score and 0.921 accuracy it outperforms BERT. RoBERTa manages an accuracy of 0.937% which is better than both BERT and XLNet, however, it is far lower than the proposed model.

## Performance comparison with additional models

For showing the superior performance of the proposed model, this study selected several models reported in [62] and in [77]. The complete comparison in terms of accuracy, f-score, f-agree, f-disagree, f-discuss, and f-unrelated is shown in Table 6. Results show that the proposed ensemble model shows far better results than these models and obtains the highest accuracy and F-score values both average F-score and individual F-score for each class. The comparison of aggregated FNC Score with research published in 2023 is shown in Table 7.

**Table 6. Performance comparison of FakeNET model with state-of-the-art approaches.**

| Model Name | A | F | F-agree | F-disagree | F-discuss | F-unrelated |
|---|---|---|---|---|---|---|
| TalosComb [62] | 0.820 | 0.582 | 0.539 | 0.035 | 0.760 | 0.994 |
| TalosTree [62] | 0.830 | 0.570 | 0.520 | 0.003 | 0.762 | 0.994 |
| TalosCNN [62] | 0.502 | 0.308 | 0.258 | 0.092 | 0.0 | 0.882 |
| Athene [62] | 0.820 | 0.604 | 0.487 | 0.151 | 0.780 | 0.996 |
| UCLMR [62] | 0.817 | 0.583 | 0.479 | 0.114 | 0.747 | 0.989 |
| featMLP [62] | 0.825 | 0.607 | 0.530 | 0.151 | 0.766 | 0.982 |
| stackLSTM [62] | 0.821 | 0.609 | 0.501 | 0.180 | 0.757 | 0.995 |
| Upperbound [62] | 0.859 | 0.754 | 0.588 | 0.667 | 0.765 | 0.997 |
| Base [77] | 0.871 | 0.785 | 0.502 | 0.272 | 0.724 | 0.965 |
| Inf1 [77] | 0.869 | 0.778 | 0.506 | 0.278 | 0.717 | 0.963 |
| Inf3 [77] | 0.870 | 0.781 | 0.506 | 0.288 | 0.722 | 0.963 |
| BERT1 [77] | 0.875 | 0.790 | 0.537 | 0.314 | 0.712 | 0.968 |
| BERT3 [77] | 0.877 | 0.794 | 0.531 | 0.340 | 0.716 | 0.970 |
| BERT3 + Inf3 [77] | 0.879 | 0.797 | 0.535 | 0.312 | 0.738 | 0.969 |
| BERT [77] | 0.814 | 0.686 | 0.431 | 0.317 | 0.585 | 0.916 |
| featMLP [77] | 0.871 | 0.785 | 0.502 | 0.272 | 0.724 | 0.965 |
| BERTOptimized [77] | 0.913 | 0.861 | 0.647 | 0.478 | 0.800 | 0.986 |
| XLNet [77] | 0.921 | 0.879 | 0.686 | 0.548 | 0.821 | 0.986 |
| RoBERTa [77] | 0.931 | 0.891 | 0.707 | 0.580 | 0.845 | 0.991 |
| **FakeNET** | 0.978 | 0.978 | 0.860 | 0.760 | 0.880 | 0.990 |

**Table 7. Aggregated FNC-score comparison with research published in 2023.**

| Model | FNC Score |
|---|---|
| GB Classifier [80] | 0.795 |
| CNN + GBDT [80] | 0.820 |
| CS [80] | 0.890 |
| ES + LSTM [80] | 0.901 |
| ES + LSTM + AT [80] | 0.891 |
| ES + LSTM + AT (optimized) [80] | 0.934 |
| **FakeNET** | 0.978 |

## Results of cross-validation

A 10-fold cross-validation is also done to validate the performance of the proposed approach and results are presented in Table 8. It can be observed that the proposed model provides an average accuracy of 97.1% while the average values for P, R, and F are 96.9%, 98.0%, and 97.4%, respectively.

## Discussions

This study presents a model for fake news stance detection which combines CNN and LSTM to make an ensemble model. The model is trained on features selected by the PCA algorithm. Minimal processing power and storage space are needed. Additionally, a lot of calculation is not necessary [34]. The benefits in terms of both time and space complexity are substantial. The performance of the proposed model is compared with several other models. While RoBERTa's 125 million parameters all need to be fine-tuned, the proposed model just use a small set of parameter to produce better results. Furthermore, the computational complexity of RoBERTa is very high when 12 layers and 768 units are used. Similarly, BERT and XLNet show poor results, especially when F-score is considered. Using dimensionality reduction techniques also shortens the amount of time needed for training. The proposed model is good both in terms of accuracy and computational complexity.

Despite their usefulness, feature reduction strategies have some limitations as well. To get better results, the features in the dataset should be co-related. Additionally, this study is restricted to the English language only, and employing it in any other language having differences in writing style may produce very different results.

**Table 8. Significance of FakeNET with k-fold validation.**

| K-folds | A | P | R | F |
|---|---|---|---|---|
| 1st-Fold | 0.974 | 0.974 | 0.984 | 0.979 |
| 2nd-Fold | 0.978 | 0.962 | 0.965 | 0.963 |
| 3rd-Fold | 0.972 | 0.963 | 0.986 | 0.974 |
| 4th-Fold | 0.971 | 0.964 | 0.974 | 0.969 |
| 5th-Fold | 0.978 | 0.979 | 0.992 | 0.985 |
| 6th-Fold | 0.959 | 0.963 | 0.971 | 0.967 |
| 7th-Fold | 0.969 | 0.968 | 0.974 | 0.971 |
| 8th-Fold | 0.979 | 0.975 | 0.992 | 0.983 |
| 9th-Fold | 0.964 | 0.969 | 0.981 | 0.975 |
| 10th-Fold | 0.972 | 0.973 | 0.981 | 0.977 |
| **Average** | **0.971** | **0.969** | **0.980** | **0.974** |

## Conclusion and Future Work

Detecting false news is critical for online platforms, where a large amount of changed and engineered content is published every day. Based on the body of the news and headlines, this study proposes an ensemble approach for detecting false news. For this task, the proposed model utilizes CNN and LSTM while the PCA technique is employed to extract appropriate features. To analyze the model's performance, various experiments are carried out using preprocessing, no preprocessing, and features based on PCA and Chi-Square. Results demonstrate that the use of data preprocessing leads to substantial improvement in the results. Both PCA and Chi-Square tend to improve the results compared to the use of full features. Results from the proposed method (FakeNET + PCA) are superior with an accuracy of 97.8% which is significantly higher than BERT, XLNet, and RoBERTa models. When compared to existing state-of-the-art methodologies, the proposed model gives better results. Furthermore, k-fold cross-validation also shows the robustness of the model. In the future, we plan to test the model on large and complex datasets. We also intend to investigate how an ensemble of the machine and deep learning models can improve overall performance.

## Author Contributions

**Conceptualization:** Xiaochun Cheng, Imran Ashraf.

**Data curation:** Turki Aljrees, Khaled Alnowaiser.

**Funding acquisition:** Turki Aljrees, Xiaochun Cheng.

**Investigation:** Rizwan Majeed, Nihal Abuzinadah.

**Methodology:** Muhammad Umer, Nihal Abuzinadah.

**Project administration:** Turki Aljrees, Nihal Abuzinadah.

**Software:** Turki Aljrees, Mian Muhammad Ahmed, Muhammad Umer, Rizwan Majeed.

**Supervision:** Xiaochun Cheng, Khaled Alnowaiser, Imran Ashraf.

**Validation:** Rizwan Majeed, Khaled Alnowaiser.

**Visualization:** Mian Muhammad Ahmed.

**Writing – original draft:** Muhammad Umer.

**Writing – review & editing:** Rizwan Majeed, Imran Ashraf.

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
