## [Decision Letter · Decision Letter 0]

3 Apr 2023

PONE-D-23-02497Fake News Stance Detection Using Selective Features and FakeNETPLOS ONE

Dear Dr. Umer,

Thank you for submitting your manuscript to PLOS ONE. After careful consideration, we feel that it has merit but does not fully meet PLOS ONE’s publication criteria as it currently stands. Therefore, we invite you to submit a revised version of the manuscript that addresses the points raised during the review process.

ACADEMIC EDITOR: When updating your manuscript, you should elaborate on your points and clarify with references, examples, data, etc. Also, note that if a reviewer suggested references, you should only add those that are relevant to your work if you feel they strengthen your article. Recommending references to specific publications is not appropriate for reviewers.

We look forward to receiving your revised manuscript.

Kind regards,

Mohamed Hammad, Ph.D.

Academic Editor

PLOS ONE

3. In your Methods section, please include additional information about your dataset and ensure that you have included a statement specifying whether the collection and analysis method complied with the terms and conditions for the source of the data.

Reviewers' comments:

Reviewer's Responses to Questions

**Comments to the Author**

1. Is the manuscript technically sound, and do the data support the conclusions?

Reviewer #1: Yes

Reviewer #2: Partly

Reviewer #3: Yes

2. Has the statistical analysis been performed appropriately and rigorously? 

Reviewer #1: Yes

Reviewer #2: Yes

Reviewer #3: Yes

3. Have the authors made all data underlying the findings in their manuscript fully available?

Reviewer #1: Yes

Reviewer #2: Yes

Reviewer #3: Yes

4. Is the manuscript presented in an intelligible fashion and written in standard English?

Reviewer #1: No

Reviewer #2: Yes

Reviewer #3: Yes

5. Review Comments to the Author

Reviewer #1: This paper presents a fake news detection using FakeNET model. The proposed model is a combination of both CNN with LSTM. The basic idea sounds good but I have some comments to improve the readability of the work as follows:

1. Introduction section is not well organized. Long paragraphs should be divided into short.

2. Comparison with state of the art techniques required.

3. Include specific author contributions at the end of the introduction section.

4. Refer the following important works on importance of deep learning models

https://doi.org/10.1016/j.bspc.2022.104549

https://doi.org/10.3390/su15010133

https://doi.org/10.3390/info14020065

https://doi.org/10.1016/j.engappai.2022.105731

5. In Line 177, The extracted features are fed into the proposed hybrid model which comprises CNN and LSTM” I didn’t understand why authors fed selected features to deep learning, because DL has inbuilt feature learning approach?

6. How the data preprocessing unit work? it is not clear.

7. Elaborate Performance evaluation metrics with equations.

8. Tuning of the hyperparameters not discussed well.

9. Cite the SOTA models in Table 4.

Overall idea was good, results also reliable but needs improvement in the structure of the script.

Reviewer #2: Dear Authors

I have now completed the review of the manuscript titled " Fake News Stance Detection Using Selective Features and FakeNET”. The purpose of this study is to locate the article's perspective concerning the headline. This study employs two methods for reducing the number of dimensions including Chi-square and principal component analysis (PCA) in conjunction with a hybrid neural network architecture that combines a convolutional neural network (CNN) and long short-term memory (LSTM) model called FakeNET. Dimensionality reduction strategies are used in this work to utilize appropriate feature vectors for better performance and lower computational complexity. A multi-class dataset is used comprising 'agree', 'disagree', 'discuss', and 'unrelated' classes obtained from the Fake News Challenges (FNC) website.

1. Authors should edit the manuscript in English.

2. The Abstract has a lack of flow.

3. The purpose of the study should come early in the abstract.

4. How authors addressed the issues while using PCA including PCA assumption of linear relationships, outliers treatment; normality; interpretation of components and how to select the optimized number of components.

5. Proposed Model: “Filters f with varying kernel size The output is calculated as (c × e),

where e is a word embedding, and c is applied to each e.”, please clarify this sentence.

6. Proposed Model: The FakeNET model comprises a convolutional layer with 64 filters of 5×5 size, why not 3X3 and why not 32 filters?

7. It is followed by the max pooling layer with a 4-pool size, how the authors have chosen this number?

8. Each LSTM layer has 100 neurons while the dense layer has 50 neurons, why?

9. The dropout layer is used with a 0.2 dropout rate, why not 0.4 or 0.5?

10. The LSTM layer is used with 100 units, why not 50 or 200?

11. This study used rectified linear unit (ReLU) as the activation, The ReLU activation function has several problems, including vanishing gradients, dead neurons, lack of smoothness, not centred around zero, and output range limitation. How authors addressed these problems?

12. Line 153-173: LSTM and CNN are used in the present investigation. Authors should use the latest research using LSTM and CNN in various applications including cyber security.

13. The introduction section requires some relevant articles which emphasize the need and use of LSTM and CNN for various types of applications. The authors should add the following works as planetscope nanosatellites image classification using machine learning; cnn based automated weed detection system using uav imagery; smotednn: a novel model for air pollution forecasting and aqi classification; cdlstm: a novel model for climate change forecasting; deep learning based modeling of groundwater storage change; deep learning based supervised image classification using uav images for forest areas classification; insider threat detection based on nlp word embedding and machine learning; dbotpm: a deep neural network-based botnet prediction model; dnnbot: deep neural network-based botnet detection and classification; development of pccnn-based network intrusion detection system for edge computing, to set the narative of the CNN and LSTM utility and wider acceptability in various areas

14. Performance Evaluation Metrics, please write the formulas, and justify if the authors used an F-1 score, F-0.5 or F-2 and why?.

15. Add the future scope of the present work.

16. Discuss the limitations of the proposed research.

17. The conclusion should be rephrased with more clarity and major takeaways from the work.

Reviewer #3: This study presents FakeNET, a hybrid deep-learning model that ensembles Convolutional Neural Networks (CNN) and long short-term memory (LSTM) with both Principal Component Analysis (PCA ) and Chi-square feature reduction approaches. The result shows that FakeNet achieves excellent accuracy, Precision, recall, and F1 score when trained with (PCA) based features. Also, in comparison with other deep learning FakeNet achieves good complexity success. This is good research work and is technically sound. However, there is minor work to be made, such as the update of the references to reflect the 2021-2023 published works on fake news detection.

6. PLOS authors have the option to publish the peer review history of their article (what does this mean?). If published, this will include your full peer review and any attached files.

Reviewer #1: **Yes: **Dr Kiran Kumar Patro

Reviewer #2: No

Reviewer #3: No

---

## [Author Response · Author response to Decision Letter 0]

7 May 2023

We have uploaded a separate file in submission files to respond to reviewer comments.

---

## [Decision Letter · Decision Letter 1]

4 Jun 2023

Fake News Stance Detection Using Selective Features and FakeNET

PONE-D-23-02497R1

Dear Dr. Umer,

We’re pleased to inform you that your manuscript has been judged scientifically suitable for publication and will be formally accepted for publication once it meets all outstanding technical requirements.

Kind regards,

Mohamed Hammad, Ph.D.

Academic Editor

PLOS ONE

Additional Editor Comments (optional):

Reviewers' comments:

Reviewer's Responses to Questions

**Comments to the Author**

1. If the authors have adequately addressed your comments raised in a previous round of review and you feel that this manuscript is now acceptable for publication, you may indicate that here to bypass the “Comments to the Author” section, enter your conflict of interest statement in the “Confidential to Editor” section, and submit your "Accept" recommendation.

Reviewer #2: All comments have been addressed

Reviewer #3: All comments have been addressed

2. Is the manuscript technically sound, and do the data support the conclusions?

Reviewer #2: Yes

Reviewer #3: Yes

3. Has the statistical analysis been performed appropriately and rigorously? 

Reviewer #2: Yes

Reviewer #3: Yes

4. Have the authors made all data underlying the findings in their manuscript fully available?

Reviewer #2: Yes

Reviewer #3: Yes

5. Is the manuscript presented in an intelligible fashion and written in standard English?

Reviewer #2: Yes

Reviewer #3: Yes

6. Review Comments to the Author

Reviewer #2: The revised version of the manuscript entitled "Fake News Stance Detection Using Selective Features and FakeNET", have addressed all the comments satisfactorily.

Reviewer #3: he authors have successfully addressed my comments. The introduction section of the work has been properly written with changes to include major elements and recent related works have been incorporated in the study. The justification for the use of PCA was clearly stated and the performance comparison of the proposed model with the existing models were performed excellently. There are significant improvements in this manuscript as compared to the previous version. I am satisfied with the work.

7. PLOS authors have the option to publish the peer review history of their article (what does this mean?). If published, this will include your full peer review and any attached files.

Reviewer #2: No

Reviewer #3: No

---

## [Editor Report · Acceptance letter]

21 Jul 2023

PONE-D-23-02497R1 

Fake News Stance Detection Using Selective Features and FakeNET 

Dear Dr. Umer:

I'm pleased to inform you that your manuscript has been deemed suitable for publication in PLOS ONE. Congratulations! Your manuscript is now with our production department. 

Kind regards, 

on behalf of

Dr. Mohamed Hammad 

Academic Editor

PLOS ONE